

# Lessepsian migration and parasitism: richness, prevalence and intensity of parasites in the invasive fish *Sphyraena chrysotaenia* compared to its native congener *Sphyraena sphyraena* in Tunisian coastal waters

Wiem Boussellaa[1,2], Lassad Neifar[1], M. Anouk Goedknegt[2] and David W. Thieltges[2]

[1] Department of Life Sciences, Faculty of Sciences of Sfax, Sfax University, Sfax, Tunisia
[2] Department of Coastal Systems, NIOZ Royal Netherlands Institute for Sea Research and Utrecht University, Den Burg Texel, Netherlands

## ABSTRACT

**Background**. Parasites can play various roles in the invasion of non-native species, but these are still understudied in marine ecosystems. This also applies to invasions from the Red Sea to the Mediterranean Sea via the Suez Canal, the so-called Lessepsian migration. In this study, we investigated the role of parasites in the invasion of the Lessepsian migrant *Sphyraena chrysotaenia* in the Tunisian Mediterranean Sea.

**Methods**. We compared metazoan parasite richness, prevalence and intensity of *S. chrysotaenia* (Perciformes: Sphyraenidae) with infections in its native congener *Sphyraena sphyraena* by sampling these fish species at seven locations along the Tunisian coast. Additionally, we reviewed the literature to identify native and invasive parasite species recorded in these two hosts.

**Results**. Our results suggest the loss of at least two parasite species of the invasive fish. At the same time, the Lessepsian migrant has co-introduced three parasite species during the initial migration to the Mediterranean Sea, that are assumed to originate from the Red Sea of which only one parasite species has been reported during the spread to Tunisian waters. In addition, we found that the invasive fish has acquired six parasite species that are native in the Mediterranean Sea. However, parasite richness, prevalence and intensity were overall much lower in the invasive compared to the native fish host in the Mediterranean Sea.

**Discussion**. These results suggest that the Lessepsian migrant may affect native fish hosts by potentially altering the dynamics of native and invasive parasite-host interactions via parasite release, parasite co-introduction and parasite acquisition. They further suggest that the lower infection levels in the invasive fish may result in a competitive advantage over native fish hosts (enemy release hypothesis). This study demonstrates that cross-species comparisons of parasite infection levels are a valuable tool to identify the different roles of parasites in the course of Lessepsian migrations.

**Subjects** Biodiversity, Ecology, Marine Biology, Parasitology, Taxonomy

Corresponding author
Wiem Boussellaa,
wiem.boussellaa@hotmail.com

**Keywords** Enemy release, Parasite co-introduction, Parasite spillback, Mediterranean Sea, Parasite spillover, Red Sea

## INTRODUCTION

One of the potential explanations for the establishment and subsequent spread of invasive species in marine and other ecosystems is the enemy release hypothesis (*Elton, 1958*; *Keane & Crawley, 2002*). This hypothesis states that invasive species may gain a competitive advantage over native species by losing all or part of their natural enemies, such as predators and parasites, during the invasion process. For parasites, such a release or reduction has been documented for a wide range of host taxa including marine species (*Torchin, Lafferty & Kuris, 2001*; *Keane & Crawley, 2002*; *Torchin et al., 2003*; *Torchin & Mitchell, 2004*; *Blakeslee, Fowler & Keogh, 2013*). However, invasive hosts do not necessarily lose all their native parasites during an invasion, but can often co-introduce parasites to their invaded range (*Lymbery et al., 2014*). The likelihood of this co-introduction depends, among others, on the host specificity and life cycle of the respective parasite species. Generalist parasites which infect a larger range of host species and parasites with direct life cycles are more likely to be co-introduced than highly specific parasites or parasites with complex life cycles (i.e., depending on several sequential different host species; *Torchin, Lafferty & Kuris, 2002*; *Poulin & Morand, 2004*; *Lymbery et al., 2014*). In the new range, co-introduced parasites may also infect native hosts (parasite spillover; *Prenter et al., 2004*; *Kelly et al., 2009*), with potentially serious impacts on native species and ecosystems (emerging diseases; *Daszak, Cunningham & Hyatt, 2000*). Finally, invasive species can also acquire native parasites from native host species. This parasite acquisition may have deleterious effects on naïve invasive host species (increased susceptibility hypothesis *sensu Colautti et al., 2004*), but may also ultimately amplify native parasite population sizes, resulting in increased parasite loads in native hosts, a phenomenon which is referred to as parasite spillback (*Kelly et al., 2009*). While many of these mechanisms may result in a competitive advantage for invasive over native species, the magnitude of this advantage will depend on the actual difference in parasite infection levels between invasive and native hosts. Several studies comparing infection levels in invasive and introduced hosts (community studies or cross-species comparisons; *sensu Colautti et al., 2004*; *Torchin & Mitchell, 2004*) have shown that infection levels are often lower in invasive host species (*Georgiev et al., 2007*; *Dang et al., 2009*; *Roche et al., 2010*; *Gendron, Marcogliese & Thomas, 2012*). However, such cross-species comparisons of infection levels in invasive and native competitors are surprisingly scarce (*Goedknegt et al., 2017*). In addition, our knowledge on the role of parasites in biological invasions in general is still limited, especially in the marine realm (*Vignon & Sasal, 2010*; *Goedknegt et al., 2016*).

This also applies to species invasions in the Mediterranean Sea, an ecosystem with an extraordinarily high rate of species introductions, with more than 1,000 alien species listed (*Bilecenoglu et al., 2013*). Especially via the Suez Canal, which was opened in 1869, many benthic invertebrates and fish species have migrated from the Indian Ocean via the Red Sea to the Mediterranean Sea, a massive human-initiated invasion referred to as

Lessepsian migration *sensu Por, 1978*. Despite this high migration rate and the resulting high number of species introductions, parasitological investigations of invasive species have been surprisingly rare in this region (*Pérez-del Olmo, Kostadinova & Gibson, 2016*) and have only focused on a few host species such as a portunid crab (*Galil & Innocenti, 1999*), the Lessepsian fishes *Siganus* spp. (*Diamant, 2010*), *Fistularia commersonii* (*Merella et al., 2016*), *Etrumeus golanii* (*Boussellaa et al., 2016*) and *Lagocephalus sceleratus* (*Bakopoulos, Karoubali & Diakou, 2017*). Initially, 18 species of parasites spread over four taxonomic groups (Monogenea, Crustacea, Protozoa and Digenea) had been recognized as Lessepsian migrants which have been co-introduced with their hosts to the Mediterranean Sea (*Zenetos et al., 2008*), and more recently this list has been updated by the addition of many other parasite species (*Diamant, 2010*; *Merella et al., 2016*). Given that there are many Lessepsian migrants that have never been under parasitological investigation, the actual list of co-introduced parasite species is likely to be much longer. In addition, invasive hosts may have acquired native parasite species but whether the resulting parasite loads of hosts are actually lower, equal to or higher than the ones in native species in the Mediterranean Sea has scarcely been studied to date.

In the present study, we investigated metazoan parasite infections in the Lessepsian migrant fish *Sphyraena chrysotaenia* Klunzinger, 1884, along the Tunisian coast in the central Mediterranean Sea. This fish species was first recorded in the eastern Mediterranean Sea in 1931 (*Spicer, 1931*) and has since then spread westwards, after which it was reported for the first time at the Tunisian coast in 2002 (*Bradai et al., 2002*). It is now a relatively common piscivorous fish of about 20–25 cm length, living in the pelagic and demersal zones to a depth of 50 m in inshore waters where it is captured by local artisanal fisheries (*Golani & Ben Tuvia, 1995*; *Wadie & Riskallah, 2001*; *Zouari-Ktari, Bradai & Bouain, 2009*). A cross-species comparison of the parasite communities was carried out considering the parasite richness and levels of infection in the invasive fish and its native congeneric *Sphyraena sphyraena* Linnaeus, 1758. This native fish species is usually larger than the invasive species (30–60 cm length) but has an overlapping prey spectrum (*Kalogirou et al., 2012*), lives in the same habitats and is also used by local fisheries (*Relini & Orsi Relini, 1997*; *Allam, Faltas & Ragheb, 2005*). By sampling invasive (*S. chrysotaenia*) and native (*S. sphyraena*) fish hosts along the Tunisian coast and by conducting an additional parasitological literature survey, we aimed to answer the following specific research questions: (1) is there evidence that *S. chrysotaenia* experienced a release from its native parasites from the Red Sea?, (2) did *S. chrysotaenia* co-introduce parasites from the Red Sea and/or did it acquire native parasites from the Mediterranean Sea? and (3) how do parasite richness, prevalence and intensity in the invasive *S. chrysotaenia* compare with those in the native *S. sphyraena*?

## MATERIAL AND METHODS

### Fish sampling
Between October 2012 and July 2015, a total of 107 specimens of *S. sphyraena* (native fish) and 148 specimens of *S. chrysotaenia* (invasive fish) were collected at seven fishing localities along the Tunisian coast (off the cities of Sfax, Kerkennah, Skhira, Chebba, Zarat, Zarzis

**Table 1  Information on the sampling design of fish hosts collected for this study in Tunisian coastal waters.** Given are the location numbers used in Fig. 1, location name, geographic coordinates, sampling dates and sample sizes (per sex) for the two fish host species (*S. sphyraena* and *S. chrysotaenia*).

| Location number | Location name | Geographic coordinates | Sampling dates | *Sphyraena sphyraena* (native) | | *Sphyraena chrysotaenia* (invasive) | |
|---|---|---|---|---|---|---|---|
| | | | | Females | Males | Females | Males |
| 1 | Sfax | 34°44′26″N 10°45′37″E | 22/10/2012 08/03/2013 | 3 | 3 | 9 | 9 |
| 2 | Kerkennah | 34°39′29″N 11°04′07″E | 10/02/2013 10/03/2013 10/09/2013 | 20 | 11 | 18 | 15 |
| 3 | Skhira | 34°17′57″N 10°04′11″E | 06/11/2014 04/07/2015 | 10 | 8 | 7 | 10 |
| 4 | Chebba | 35°14′14″N 11°6′54″E | 05/11/2012 | 8 | 1 | 5 | 14 |
| 5 | Zarat | 33°39′59″N 10°20′59″ | 07/02/2014 | 6 | 1 | 15 | 6 |
| 6 | Zarzis | 33°30′14″N 11°06′43″E | 11/05/2015 19/06/2015 | 11 | 6 | 9 | 7 |
| 7 | Sayada | 35°40′7″N 10°53′32″E | 07/02/2014 03/06/2014 | 15 | 4 | 7 | 17 |
| | | | | 73 | 34 | 70 | 78 |

and Sayeda; Fig. 1, Table 1). Fish were mostly bought from local fishermen operating landing trawlers along the Tunisian coast at a depth of about 30 m. In addition, some specimens were recovered from artisanal inshore fishery of Kerkennah and Chebba. Only adult individuals were examined, with total lengths of 25.7–42.7 cm for *S. sphyraena* and 18.5–26.2 cm for *S. chrysotaenia*. Samples were kept fresh or were deep-frozen in individual plastic bags at −10 °C, until further examination in the laboratory. After defrosting, all fish individuals were identified to species level using *Whitehead et al. (1984)* and *Fisher, Schneider & Bauchot (1987)* and examined for parasites as described below. The Faculty of Sciences of Sfax University provided full approval for this purely observational research.

## Parasite sampling

We focused on metazoan parasites during our study. Fish skin, fins, nasal pits, eyes and buccal cavities were thoroughly examined for the presence of ectoparasites under a stereomicroscope with incident light. Gill arches were separated by incision, placed in petri dishes filled with sea water and examined for the presence of ectoparasites. Internal organs (stomach, pyloric caeca, intestines, heart, liver, spleen, gall bladder and gonads) were separated and individually examined for the presence of endoparasites. Platyhelminthes were fixed between a slide and coverslip with 70% ethanol. Fixed specimens were stained with Semichon's acetic carmine, dehydrated using a graded ethanol series then cleared in clove oil and mounted in Canada balsam. Other parasites (copepods, isopods, nematodes and annelids) were directly fixed in 70% ethanol for later examination. Parasites were

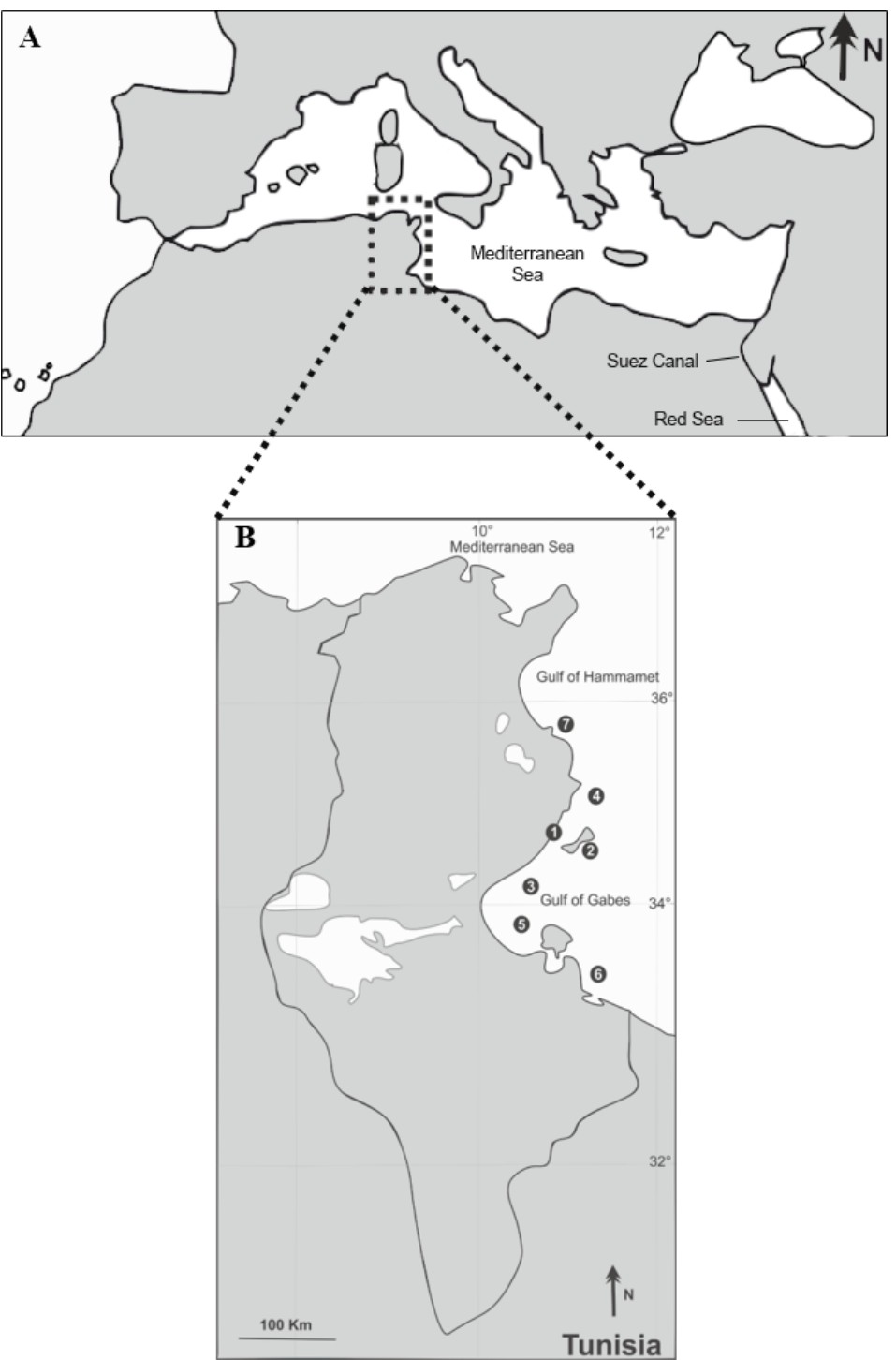

**Figure 1 Sampling locations (1–7) of the native *S. sphyraena* and the invasive Lessepsian migrant *S. chrysotaenia* in the central Mediterranean Sea (A) along the Tunisian coast (B).** For location names, coordinates, sampling dates and sampling effort per species see Table 1.

identified to the lowest taxonomic level possible using *Gibson, Jones & Bray (2002)* for Digenea, *Neifar (1995)* and *Theisen et al. (2017)* for Monogenea, *Euzet (1994)* for Cestoda, *Berland (1961)* and *Petter & Maillard (1988)* for Nematoda and the keys of *Kabata (2003)* for Copepoda.

## Literature review

We searched literature databases (the Host-Parasite database from the Natural History Museum in London, Web of Science and Google Scholar) for published records of additional parasite species from the Mediterranean Sea and the Red Sea. Search strings included the species names and the different parasite taxa. In addition, we searched the reference sections of publications and our own reference collections for potential further studies reporting parasite infections in the two fish.

## Statistical analyses

As sampling effort differed between native ($n = 107$) and invasive fish hosts ($n = 148$), we produced rarefaction curves to identify the level of dependence of species accumulation on sampling effort. Sample-based rarefaction curves were computed based on sample sizes at the different locations (*Gotelli & Colwell, 2001*) using Estimates 9.1.0 (*Colwell, 2013*). Based upon recommendations by *Walther & Morand (1998)* in regard to parasitological research, we used the nonparametric species estimator Chao2 for our rarefaction analyses. This species estimator algorithm uses the frequency of unique species in samples to estimate the number of missing species in a population (*Chao, 2005*).

We used general linear models (GLMs) to test for statistical differences between the fish species and among locations in parasite richness of individual fish (Poisson distribution; $n = 255$), infection status of individual fish (infected or uninfected; binomial distribution; $n = 255$) and parasite intensity of infected fish (negative binomial distribution; $n = 127$). In each model, we added host species and location as fixed factors and an interaction term. As the parasite fauna of the two host species was very different, we used the total infection status and intensity of parasite species per host species to compare infection levels between hosts. Although this procedure may obscure the potentially different effects on hosts exerted by different parasite species, this lumping procedure still allows for an approximate comparison of overall infection levels. We did not add fish size as a covariate in the models because preliminary analyses using GLMs did not show an effect of fish size on any of the response variables. All statistical models were run using the statistical software environment R v3.3.0. (*R Development Core Team, 2016*).

## RESULTS

After the dissections of 107 individuals of the native *S. sphyraena* and 148 individuals of the invasive *S. chrysotaenia* caught along the coast of Tunisia, ten different parasite species infecting the two fish were found. Our additional literature survey added another ten parasite species records from the Mediterranean or Red Sea to the total parasite species list of both fish species (Table 2).

Boussellaa et al. (2018), *PeerJ*, DOI 10.7717/peerj.5558

Peer J

**Table 2  Parasite species of *S. sphyraena* and *S. chrysoteania* found in this study and recorded in the literature.** For each species, the type of life cycle, host specificity, the inhabitant status in the Mediterranean Sea and the occurrence in the two host species and region (Mediterranean Sea and Red Sea) are given. If quantitative data were available, mean prevalence and intensity (± SE) in a host species and region are given. + denotes published records of specific parasite species in a host species in a region; (+) denotes an assumed occurrence in a host species and region without published records.

| Parasite taxa | Parasite species | Life cycle | Host specificity | Status Med. Sea | *Sphyraena sphyraena* (native range-Med. Sea) | *Sphyraena chrysotaenia* (invasive range - Med. Sea) | *Sphyraena chrysotaenia* (native range - k Red Sea) | Reference |
|---|---|---|---|---|---|---|---|---|
| **Annelida** | *Piscicolid* sp. | complex | generalist | native | | 0.7% (1 ± 0) | | This study |
| **Cestoda** | Tetraphyllidea | complex | generalist | native | 3.7% (16 ± 0.4) | | | This study |
| **Copepoda** | *Caligus* sp. | direct | Specialist/ generalist? | native | | 3.8% (1 ± 0) | | This study |
| | *Bomolochus unicirrus* | direct | specialist | native | 29% (2 ± 0.1) | | | This study |
| | *Nothobomolochus denticulatus* | direct | specialist | invasive | | + | (+) | *El-Rashidy & Boxshall (2012)* |
| | *Pennella filosa* | complex | generalist | native | + | | | *Ramdane, Bensouilah & Trilles (2009)* |
| **Digenea** | *Lecithochirium* sp. | complex | generalist | native | | 12.2% (1.4 ± 0) | | This study |
| | *Didymozoon sphyraenae* | complex | specialist | native | 85% (5.5 ± 0.4) | | | This study |
| | *Bucephalus sphyraenae* | complex | specialist | not present | | | + | *Nahhas, Sey & Nakahara (2006)* |
| | *Bucephalus labracis* | complex | specialist | native | | + | | *Fischthal (1982)* |
| | *Plerurus digitatus* | complex | generalist | native | + | | | *Looss (1899)* |
| **Isopoda** | *Gnathia* sp. | complex | generalist | native | 0.9% (13 ± 0.1) | 0.7% (2 ± 0) | | This study |
| | *Cymothoa indica* | complex | generalist | invasive | | + | (+) | *Trilles & Bariche (2006)* |
| | *Anilocra physodes* | complex | generalist | native | | + | | *İnnal, Kirkim & Erk'akan (2007)* |
| **Monogenea** | *Pseudempleurosoma* sp. | direct | specialist | invasive | | 4 % (1 ± 0) | (+) | This study |
| | *Chauhanea mediterranea* | direct | specialist | native | 5.6% (1.4 ± 0.1) | | | This study |
| | *Pseudolamellodiscus sphyraenae* | direct | specialist | not present | | | + | *Yamaguti (1953)* and *Kritsky, Jiménez-Ruiz & Sey (2000)* |
| | *Cotyloatlantica mediterranea* | direct | specialist | native | + | | | *Euzet & Trilles (1960)* |
| | *Rhinecotyle crepitacula* | direct | specialist | native | + | | | *Euzet & Trilles (1960)* |
| **Nematoda** | *Anisakis* sp. | complex | generalist | native | 1.9% (3 ± 0) | | | This study |
| **Total species richness** | | | | | **10** | **9** | **5** | |

### Parasite release

In total, five parasite species were collected from the Lessepsian migrant *S. chrysotaenia* in Tunisian coastal waters (Table 2). In addition, the literature survey indicated that four other parasite species have been found in this invasive host species elsewhere in the Mediterranean Sea (Table 2). Hence, *S. chrysotaenia* is infected by at least nine parasite species in its invaded range. In its native range, the Red Sea, two parasite species of *S. chrysotaenia* have been reported in the literature (Table 2). In addition, three of the parasite species found in *S. chrysotaenia* in the Mediterranean Sea most likely originate from the Red Sea, although published records are not available. This suggests that the Lessepsian migrant harbours at least five parasite species in its native range, the Red Sea (Table 2).

### Parasite co-introduction and acquisition

The parasitological examination and the literature survey revealed that the Lessepsian migrant *S. chrysotaenia* is infected by three parasite species in the Mediterranean Sea that are assumed to originate from the Red Sea (Table 2). Our results further revealed that the Lessepsian migrant *S. chrysotaenia* has acquired six native parasite species in the Mediterranean Sea (Table 2). Five of those species were found in our survey in Tunisian coastal waters and a fifth species has been noted in *S. chrysotaenia* elsewhere in the Mediterranean Sea (Table 2).

### Comparison with the native congeneric species *S. sphyraena*

The native congeneric species *S. sphyraena* was infected with six parasite species in Tunisian coastal waters and four additional species have been described in the literature (Table 2). Of all these parasite species, only one was shared with the introduced fish host *S. chrysotaenia*, the presumably native isopod *Gnathia* sp.. None of the parasites that were co-introduced by the Lessepsian migrant have been found to infect the native congeneric host species (Table 2).

In general, the rarefaction curves did not reach asymptotic levels in the two congeneric host species, indicating that a higher sampling effort may reveal more (albeit rare) parasite species in Tunisian coastal waters (Fig. 2). The species accumulation curves further indicate that the total parasite species richness at a given sampling effort could be higher in the native than in the invasive host species. This was also reflected in the mean parasite richness per individual fish found at the seven locations, which was generally significantly higher in the native *S. sphyraena* than in the invasive *S. chrysotaenia* (GLM: $\beta = 0.705$, SE $= 0.586$, $\Delta_{\text{Deviance}} = 110.655$, $df = 253$, $p < 0.001$; Fig. 3). However, at some locations this pattern was reversed, resulting in a significant interaction term ($\Delta_{\text{Deviance}} = 3.470$, $df = 241$, $p < 0.01$; Fig. 3).

Likewise, the infection status of fish hosts was generally significantly higher in the native *S. sphyraena* than in the invasive *S. chrysotaenia* (GLM: $\beta = 1.966$, SE $= 1.201$, $\Delta_{\text{Deviance}} = 162.599$, $df = 247$, $p < 0.001$; Fig. 4A). This was consistent over all locations as there was no significant difference in overall infection status between locations ($p = 0.294$; Fig. 4A). However, due to the absence of infections at two locations, there was a significant interaction term ($\Delta_{\text{Deviance}} = 168.19$, $df = 241$, $p < 0.05$: Fig. 4A). Mean intensity of
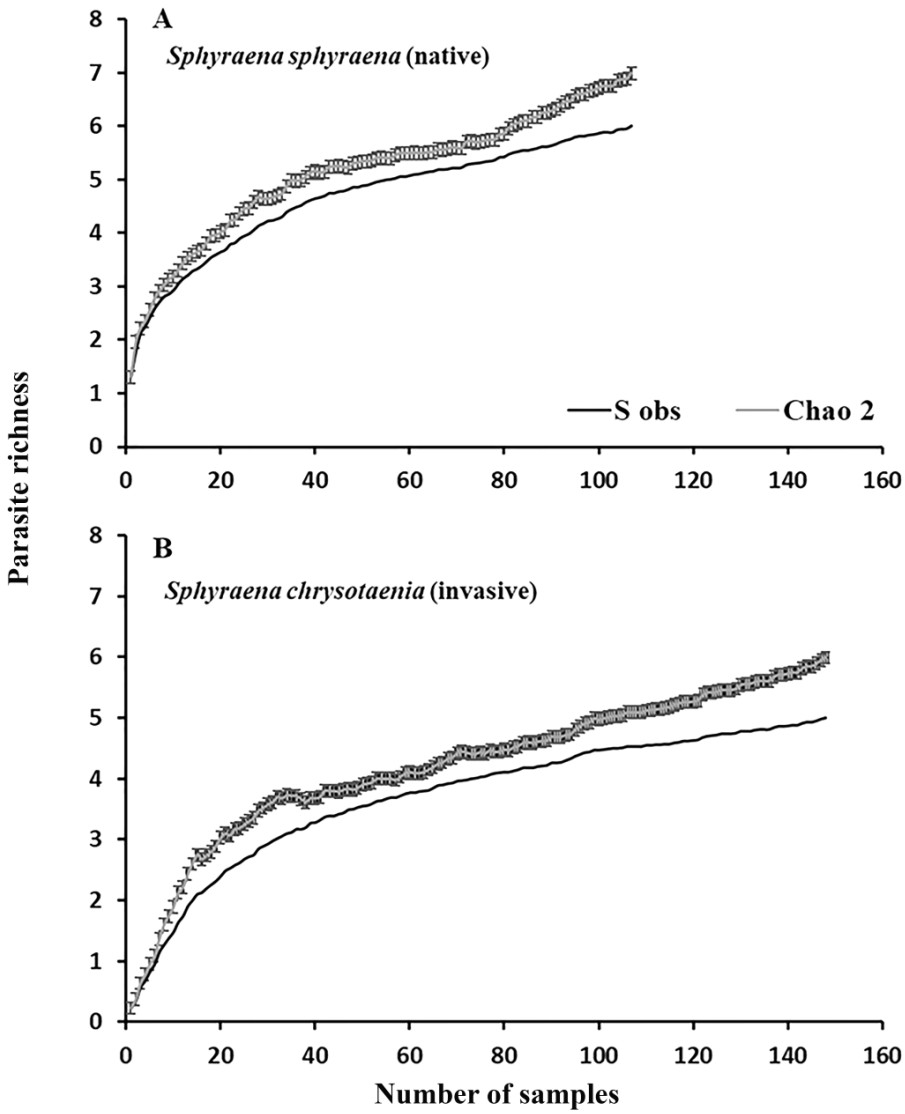

**Figure 2** Rarefraction curves for parasite richness for (A) the native fish *S. sphyraena* and (B) the invasive Lessepsian migrant *S. chrysotaenia*. Shown are the observed species richness accumulation curves ($S_{obs}$) which is the mean number of species among runs and the predicted number of species ($\pm$ SE) based on the Chao2 estimator algorithm.

infections also differed between host species, with the native fish being infected with higher numbers of parasites than the invasive fish host (GLM: $\beta = 0.337$, SE $= 0.586$, $\Delta_{\text{Deviance}} = 35.420$, $df = 119$, $p < 0.001$; Fig. 4B). There was also a significant effect of location on overall infection intensities ($\Delta_{\text{Deviance}} = 26.295$, $df = 120$, $p < 0.001$), with location 1 showing much lower values than the other locations (Fig. 4B). However, there was no significant interaction between location and host species ($p = 0.249$).

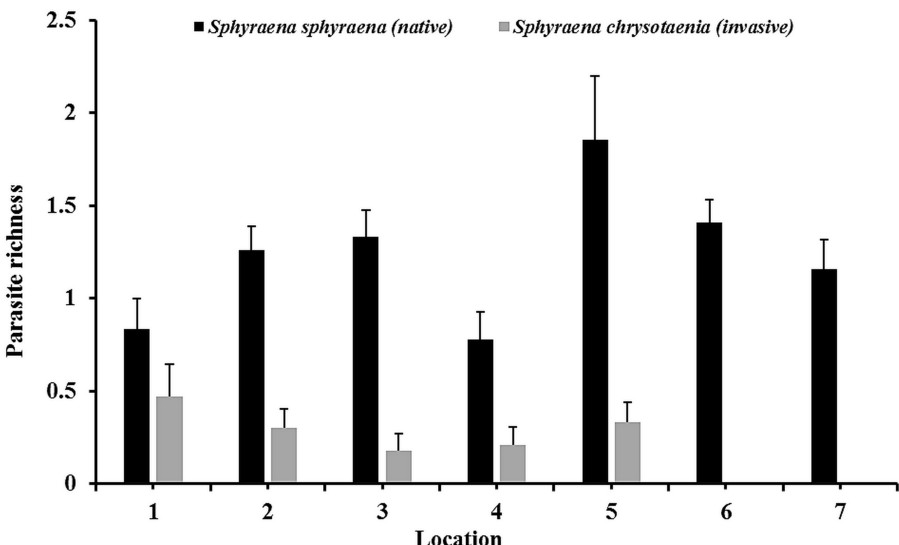

**Figure 3** **Mean parasite species richness (±SE) per individual fish.** Values are given for the at the seven locations and for both fish hosts, in the native fish *S. sphyraena* (*n* = 107) and the invasive congeneric *S. chrysotaenia* (*n* = 148).

## DISCUSSION

Based on samples from the Tunisian coast and additional literature data, our analyses indicate that the invasive Lessepsian migrant *S. chrysotaenia* lost two parasite species during its introduction to the Mediterranean Sea, but that the invasive fish also co-introduced parasites from the Red Sea and acquired one generalist parasite from native fish hosts in the Mediterranean Sea (Table 2). However, parasite richness and infection levels were overall much lower in the invasive compared to the native fish host, suggesting a potential competitive advantage for the Lessepsian migrant.

### Parasite release

In its native range (the Red Sea), the invasive *S. chrysotaenia* harbours at least five parasite species (Table 2). Of these five parasite species, only three species have been recorded in the Mediterranean Sea and only one in Tunisian coastal waters (Table 2). Hence, two parasite species, the digenean *Bucephalus sphyraenae* and the monogenean *Pseudolamellodiscus sphyraenae*, have been lost in the process of the initial invasion to the Mediterranean Sea. Two other species, the copepod *Nothobomolochus denticulatus* and the isopod *Cymothoa indica,* have been observed elsewhere in the Mediterranean Sea, but not in Tunisian coastal waters (Table 2), suggesting a loss of these parasite species in the course of the spread to the Tunisian coast. This loss of several natural parasite species is consistent with the enemy release hypothesis (*Torchin et al., 2003*; *Blakeslee, Fowler & Keogh, 2013*) and has previously been observed in other Lessepsian migrant fish, the rabbitfish *Siganus rivulatus* (see *Diamant, 1989*) and has partially been reported from the bluespotted cornet fish *Fistularia commersonii* (see *Merella et al., 2016*).

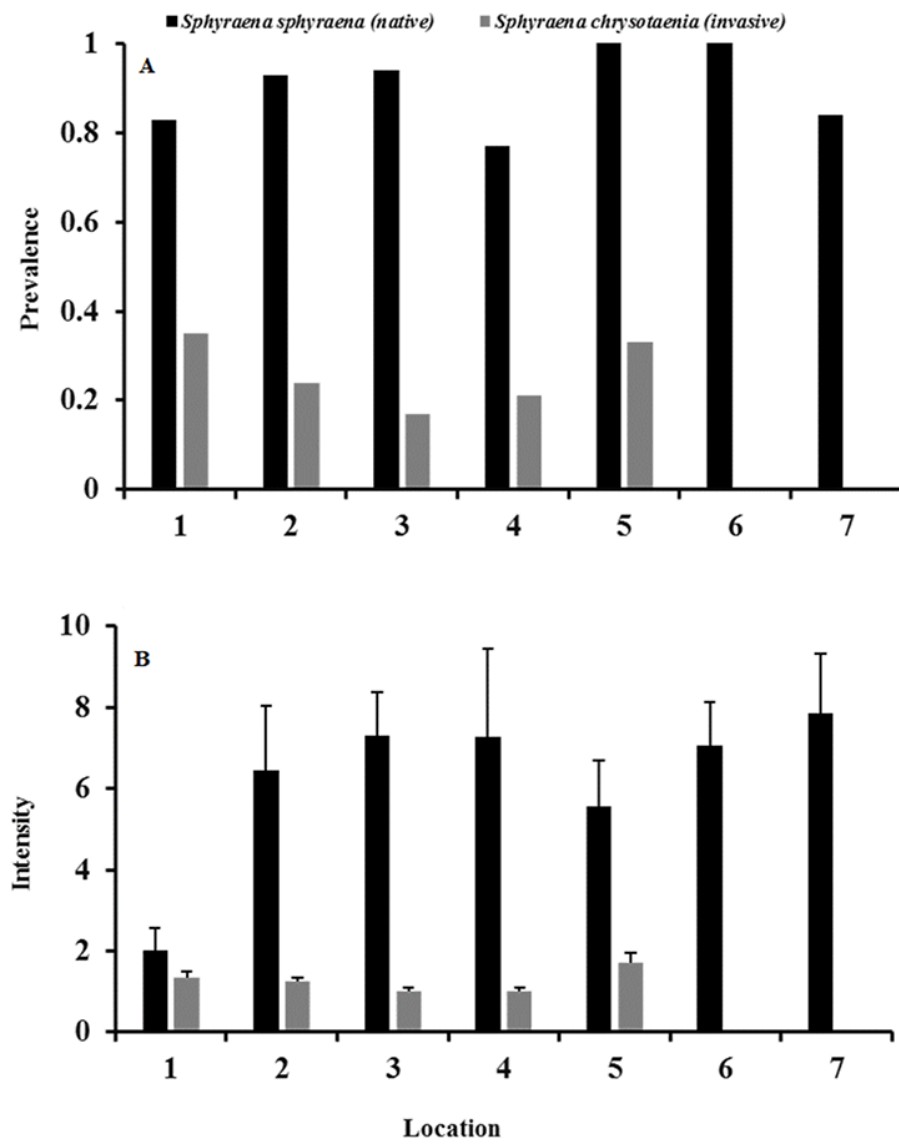

**Figure 4** **(A) Mean total parasite prevalence both fish and (B) mean intensity of total parasite loads in native *S. sphyraena* and invasive *S. chrysotaenia* at the seven sampling locations.** For sample size (*n*) per host species and location see Table 1.

Several processes may be responsible for a loss of parasites in the course of passing the Suez Canal "filter" or "bottleneck" (*Por, 1978*). First of all, parasites may not be able to cope with the environmental conditions in the canal or in the Mediterranean Sea and do not survive the migration to the new ecosystem. This negative impact of environmental conditions should be particularly relevant for ectoparasites, which are more exposed to the external environment than endoparasites, and may explain the loss of the monogenean *Pseudolamellodiscus sphyraenae*. Second, parasites may survive the passage through the canal, but, in the case of parasites with complex life cycles, they may be unable to find suitable intermediate hosts in the recipient ecosystems. In general, parasite co-introductions

seem to be more common in parasites with simple life cycles as the necessity for all hosts being present for parasites with complex life cycles makes invasions less likely (*Lymbery et al., 2014*; *Goedknegt et al., 2016*). Such a lack of suitable hosts may explain the loss of the trematode *Bucephalus sphyraenae* as trematodes are generally very host specific in respect to their first intermediate gastropod host (*Poulin & Cribb, 2002*; *Galaktionov & Dobrovolskij, 2003*). The respective gastropod host species that serves as first intermediate host for *B. sphyraenae* is not known, but a (co-)introduction of snails is unlikely because of their relatively reduced mobility. Finally, even in the case that all potential hosts are present in the new environment, the number of introduced parasite individuals may be too low to maintain a viable population. In general, propagule pressure is known to be a strong determinant of invasion success in biological invasions in general (*Wonham et al., 2000*; *Forsyth & Duncan, 2001*; *Rouget & Richardson, 2003*; *Colautti et al., 2004*). In the case of parasites, propagule pressure may be low if infection levels are low in the native region, reducing both the chances for a co-introduction and for developing viable populations after introduction to the Mediterranean Sea. This could explain the loss of infections with the monogenean *Pseudolamellodiscus sphyraenae* as this species may show only very low infection levels in its native range. These different mechanisms are not only acting during the passage of Lessepsian migrants through the Suez Canal, but also during the spread in the Mediterranean Sea after the initial introduction. This is illustrated by the copepod *Nothobomolochus denticulatus* and the isopod *Cymothoa indica* which have both been co-introduced into the Mediterranean Sea and have been found on *S. chrysotaenia* in the eastern Mediterranean Sea (*Trilles & Bariche, 2006*; *El-Rashidy & Boxshall, 2012*), but not in Tunisian waters (our study).

## Parasite co-introduction

The present results and literature review revealed that the Lessepsian migrant *S. chrysotaenia* is infected by three parasite species in the Mediterranean Sea that most likely originate from the Red Sea (Table 2). One of these species, the monogenean parasite *Pseudempleurosoma* sp., was recorded in our study in Tunisian waters. There are several reasons why we believe that this parasite species has been co-introduced to the Mediterranean Sea (although this still needs to be confirmed by records from its presumed native range).

First of all, the species has not been recorded in any native fish species in the Mediterranean Sea in earlier studies, although fish parasites are relatively well studied in this coastal ecosystem (*Oliver, 1987*; *Neifar & Euzet, 2007*; *Pérez-del Olmo, Kostadinova & Gibson, 2016*; *Chaabane et al., 2016a*; *Chaabane et al., 2016b*). Second, the genus *Pseudempleurosoma*, represented by four species in the literature, seems to have a strict specificity to its host (*Santos, Mourão & Cárdenas, 2001*), suggesting that it can only be co-introduced with its host. Finally, with their direct life cycles, monogeneans are likely to be introduced and persist in a new environment. Especially in gregarious fish such as the Sphyraenidae, the transmission of parasites with a direct life cycle can easily take place so that the lifecycle can be maintained in the new environment after an initial introduction.

The two other parasite species, the copepod *Nothobomolochus denticulatus* and the isopod *Cymothoa indica* are less host specific and have also been found on other fish

species from the Red Sea (*El-Shahawy & Desouky, 2010*). These two species have not been recorded in our study in Tunisian waters but have been reported elsewhere in the Mediterranean Sea (*Trilles & Bariche, 2006*; *El-Rashidy & Boxshall, 2012*). Both species have not been recorded in any native fish before their invasion but are considered to be co-introduced by the describing authors (*Trilles & Bariche, 2006*; *El-Rashidy & Boxshall, 2012*). However, as there are no published records of the two species available from the native range of the host, their invasive status still has to be confirmed. Both species have not (yet) been co-introduced into Tunisian coastal waters in the course of the spread of their hosts after the initial introduction in the Mediterranean Sea, probably due some of the mechanisms discussed above.

## Parasite acquisition

Besides being infected with co-introduced parasites, our study also revealed that the Lessepsian migrant *S. chrysotaenia* has acquired six native parasite species in the Mediterranean Sea. Four of those species were found in our survey in Tunisian coastal waters and a fifth species had been noted in *S. chrysotaenia* elsewhere in the Mediterranean Sea (Table 2). We consider these acquired parasite species to be native species from the Mediterranean Sea due to the following reasons. The digenean *Lecithochirium* sp. has not been reported from the invasive *S. chrysotaenia,* neither in the Mediterranean Sea nor in the Red Sea (*Fischthal, 1982*; *Nahhas, Sey & Nakahara, 2006*). Given the complex life cycle of the Hemiuridae that need at least one intermediate host species, the fact that this genus now contains at least more than 100 species (*Surekha & Lakshmi, 2005*) from which at least 10 are reported from Mediterranean Sea with scarcely reported life cycles (Table S1) and that the genus *Lecithochirium* is very specific to its first intermediate gastropod host (*Gibson & Bray, 1994*), *Lecithochirium* sp. is most likely of Mediterranean origin and has been acquired after the introduction of *S. chrysotaenia* to the Mediterranean Sea. Specimens of the copepod *Caligus* sp. reported from *S. chrysotaenia* were in poor condition and identification to the species level was impossible. According to the literature, no caligids were collected from *S. chrysotaenia* in the Red Sea but these parasitic copepods have been recorded in native Mediterranean fish (*Benmansour & Ben Hassine, 1998*; *Raibaut, Combes & Benoit, 1998*). Although many species are highly host specific there are also many *Caligus* species with low host specificity (*Yuniar, Palm & Walter, 2007*), making a host switch to the invasive host likely. However, further studies will be needed to ascertain the native status of *Caligus* individuals found on the invasive host. Annelids of the family Piscicolidae are considered to be generalists with a broad host specificity and are regularly reported from many native fish hosts from Tunisian coasts and the Mediterranean Sea in general (*Châari & Neifar, 2015*). According to the literature, no gnathiids were reported previously from *S. chrysoteania* in the Red Sea. In contrast, this isopod genus is very common in the Mediterranean Sea and along the Tunisian coasts, particularly on the skin of Sphyraenidae host species so that it is most likely native to the area. However, the native status of gnathiids found on the invader. The native Cymothoid *Anilocra physodes* has been reported from many native hosts of the Mediterranean Sea (*İnnal, Kirkim & Erk'akan, 2007*)  and was only observed from the invasive fish *S. chrysotaenia* in Turkish coastal waters (*Innal, Kirkim & Erk'akan,*

*2007*). Finally, the trematode *Bucephalus labracis*, which was not found in the Lessepsian migrant elsewhere in the Mediterranean Sea (*Fischthal, 1982*), is considered to be native as it uses a native bivalve (*Ruditapes decussatus*) as first intermediate host, a native fish (*Atherina boyeri*) as second intermediate host and a native fish (*Dicentrarchus labrax*) as definitive host (*Paggi & Orecchia, 1965*; *Maillard, 1976*; *Gargouri & Maamouri, 2005*; *Dhrif et al., 2015*). The acquisition of these native parasites may generally have potentially adverse effects naïve invasive host species, but also on native species, as the invasive additional hosts can potentially elevate infection levels in native species via parasite spill back. However, whether significant parasite spillback occurs in Tunisian waters remains to be studied.

## Cross-species comparison

Our comparison of parasite infections levels in the Lessepsian migrant with the native congeneric *S. sphyraena* revealed considerable qualitative and quantitative differences between the parasite communities of the two fish species. The native fish was infected with six parasite species in Tunisian coastal waters and three additional species have been described elsewhere in the Mediterranean Sea (Table 2). Of all these parasite species, only one was shared with the invasive fish host, the presumably native isopod *Gnathia* sp.. As only larval stages of gnathiid parasite can infect fish and all larvae are morphologically very similar, identification to species level was not possible. Native *Gnathia* species have been reported from the study area (*Châari & Neifar, 2015*) and we assume that the species found in our study is native but without further genetic work the native-invasive status in the invader cannot be fully clarified. Interestingly, none of the parasites that were co-introduced with the Lessepsian migrant has been found to infect the native congeneric host species. While the total number of parasite species found in the two fish hosts in the Mediterranean Sea was similar (nine in the invasive and ten in the native fish), the mean number of parasite species found in an individual fish was significantly higher in the native compared to the invasive fish. Although the rarefaction analyses suggested that further sampling would likely reveal more rare parasite species for each fish host, this general pattern of a lower richness of the more common parasite species in the invasive host species would remain at higher sampling efforts. In general, the parasite fauna of the invasive fish was mainly composed of generalist native parasites acquired in the new environment and one co-introduced parasite species, while the parasite community in native fish included more specialist parasites. This is consistent with the general idea that invasive hosts often lose specific parasites (especially with complex life cycles) in the course of the introduction and acquire mainly native generalist parasites in the new range (*Blakeslee, Fowler & Keogh, 2013*; *Lymbery et al., 2014*).

In addition to parasite species richness, also total parasite prevalence (parasite infection status in the models) and total infection intensity were generally higher in the native than in the invasive fish. Hence, despite the acquisition of native parasites by the invasive host *S. chrysotaenia*, the native *S. sphyraena* still showed higher total infection levels. This suggests that the invasive host is experiencing lower parasite infection levels compared with the native host species. Several factors may explain the lower infection levels of the invasive compared to the native host species. First, the relatively small size of the invasive fish

(20–25 cm length) compared to the native fish (30–60 cm) may cause a space constraint and reduce exposure of the invasive fish to native parasites. However, the lack of a significant effect of fish size on infection levels in our analyses suggests that body size may only play a minor role. Second, differences in the feeding behaviour of the two fish hosts might explain why their parasite infection levels are different. While the invasive *S. chrysotaenia* narrows its food spectrum to pelagic fish species, the native *S. sphyraena* extended their feeding to supra-benthic species (*Kalogirou et al., 2012*). The broader food spectrum of the native fish may result in a relative higher exposure to parasites. Third, host suitability of the two fish species may differ and in particular invasive hosts may be less suitable for native parasites due to compatibility issues. Finally, phylogenetic niche conservatism (*Wiens & Graham, 2005*; *Mouillot et al., 2006*) and a variety ecological factors such as microhabitat use and life history strategies (*Poulin, 2010*) may play a role.

The presumably native isopod *Gnathia* sp. was the only parasite genus shared by the two fish hosts in Tunisian coastal waters, with a higher prevalence and intensity in the native *S. sphyraena* than in the invasive *S. chrysotaenia*. However, for both fish species there was only a single record (at location 1 in the invasive host and at location 7 in the native host), compromising a formal statistical comparison, rendering a solid discussion on the subsequent host and parasite populations.

## CONCLUSION

Our study found evidence for the loss of parasite species of the Lessepsian migrant in the course of the introduction as well as for the co-introduction of parasites and the acquisition of native parasites. These results suggest that the Lessepsian migrant has the potential to affect native fish hosts by altering the population dynamics of native parasite species via parasite release, parasite co-introduction and acquisition of native parasites, resulting in increased infection levels in native hosts. They further suggest that the lower infection levels in the invasive host may give them a potential competitive advantage over native hosts. Further studies will be needed to investigate the resulting effects on native parasite dynamics and well as on native fish stocks. This study demonstrated that community studies or cross-species comparisons such as the one presented here, are valuable tools to identify the role of metazoan parasites in Lessepsian migrations.

## ACKNOWLEDGEMENTS

We thank Dr. Hela Derbel for her useful comments on parasite identification and all fishermen for their help during the fish sampling. In addition, we thank Dr. Jean-Lou Justine, Dr. Kenneth MacKenzie and one anonymous reviewer for their useful comments on an earlier draft of our manuscript.

### Funding
The University of Sfax supported the internship of W. Boussellaa at the NIOZ Royal Netherlands Institute for Sea Research. The funders had no role in study design, data collection and analysis, decision to publish, or preparation of the manuscript.

### Grant Disclosures
The following grant information was disclosed by the authors:
University of Sfax.

### Competing Interests
The authors declare there are no competing interests.

### Author Contributions
- Wiem Boussellaa and Lassad Neifar conceived and designed the experiments, performed the experiments, analyzed the data, contributed reagents/materials/analysis tools, prepared figures and/or tables, authored or reviewed drafts of the paper, approved the final draft.
- M. Anouk Goedknegt and David W. Thieltges performed the experiments, analyzed the data, prepared figures and/or tables, authored or reviewed drafts of the paper, approved the final draft.

### Animal Ethics
The following information was supplied relating to ethical approvals (i.e., approving body and any reference numbers):

The Faculty of Sciences of Sfax University provided full approval for this purely observational research.

### Data Availability
The raw data are provided in a Supplemental File.

### Supplemental Information
Supplemental information for this article can be found online at http://dx.doi.org/10.7717/peerj.5558#supplemental-information.

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
