# Peer review of "Lessepsian migration and parasitism: richness, prevalence and intensity of parasites in the invasive fish Sphyraena chrysotaenia compared to its native congener Sphyraena sphyraena in Tunisian coastal waters"

_PeerJ, doi:10.7717/peerj.5558_

## Round 0.1 · original submission · Major Revisions

I received two reviews. One is for “minor revision” and one is for “major revision”. I do not approve all modifications suggested by Reviewer #1, e.g. in my opinion the choice of the third person is a matter of style and taste and I will agree if you keep the paper as it is now for this aspect.

About the title: I believe, as one of the reviewers, that the names of the fish species (the native and the invasive species) should be indicated here.

Both reviewers had concern about the references and I believe this needs additional work. Concerning the references, here are some minor comments in addition to the reviewers’ comments:

(a) Line 316, I suppose that the purpose of these citations is to show that the parasite fauna in the South Mediterranean sea has been thoroughly studied. In this case you should delete the reference by Justine & Euzet, which is more about New Caledonia, and replace it by a few recent references in open-access peer-reviewed journals such as:
Chaabane A, Justine J-L, Gey D, Bakenhaster MD, and Neifar L. 2016. Pseudorhabdosynochus sulamericanus (Monogenea, Diplectanidae), a parasite of deep-sea groupers (Serranidae) occurs transatlantically on three congeneric hosts (Hyporthodus spp.), one from the Mediterranean Sea and two from the western Atlantic. PeerJ 4:e2233.
Chaabane A, Neifar L, Gey D, and Justine J-L. 2016. Species of Pseudorhabdosynochus (Monogenea, Diplectanidae) from groupers (Mycteroperca spp., Epinephelidae) in the Mediterranean and Eastern Atlantic Ocean, with special reference to the "beverleyburtonae group" and description of two new species. PLoS ONE 11:e0159886.
Chaabane A, Neifar L, and Justine J-L. 2015. Pseudorhabdosynochus regius n. sp. (Monogenea, Diplectanidae) from the mottled grouper Mycteroperca rubra (Teleostei) in the Mediterranean Sea and Eastern Atlantic. Parasite 22:9.
Chaabane A, Neifar L, and Justine J-L. 2017. Diplectanids from Mycteroperca spp. (Epinephelidae) in the Mediterranean Sea: Redescriptions of six species from material collected off Tunisia and Libya, proposal for the 'Pseudorhabdosynochus riouxi group’, and a taxonomic key. PLoS ONE 12:e0171392.
Moravec F, Chaabane A, Justine J-L, and Neifar L. 2016. Two gonad-infecting species of Philometra (Nematoda: Philometridae) from groupers (Serranidae) off Tunisia, with a key to Philometra species infecting serranid gonads. Parasite 23:8.

(I suspect that the second author is too modest and wants to avoid self-citation, but this is normal in this case).

(b) I have detected a few minor problems with the current list of references and I have annotated them (see attached scan). Please use the annotations for correcting the text – all this is very minor.

Reviewer 1 ·

Basic reporting

English must be improved.
Lack of a number of literature references.
Not always self-contained with relevant results to hypotheses, often verbose with inappropriate interpretations.
For details see attached pdf.

Experimental design

no comment, apart from few notes in the attached pdf.

Validity of the findings

The results paragraph, altough shows data of a certain relevance, is verbose, ad often speculative, appearing more a discussion than the results paragraph.
For details see attached pdf.

Additional comments

This is a potentially interesting study, because few data are available on the parasites of this fish in both natural habitat and invaded range, and also because there is a general lack of information on the parasites of Lessepsian fish.
Given the relevance of the parasitological data, results and the whole discussion are quite weak, and do not exploit the potential of the data.
Few comparisons with other studies on the parasites Lessepsian fish are done, and several references on are not included.
I recommend to rewrite the whole paper under the light of these considerations, taking care to write in a correct English and in the third person.
Further comments in the attached pdf.

Annotated reviews are not available for download in order to protect the identity of reviewers who chose to remain anonymous.

·

Basic reporting

The English is clear and unambiguous.
Literature references require some revision; see my comments to the authors.
The article structure is good and conforms to the usual format.
The results directly address the hypotheses to be tested.

Experimental design

The research is original, the research questions are well defined and the authors state how it fills a gap in our knowledge of the topic addressed. The study was performed to a high standard and the methods are described in detail.

Validity of the findings

The findings are valid, the data are robust and statistically sound, and the conclusions clearly stated.

Additional comments

This study makes a useful contribution to our knowledge of the role of parasites in an ecosystem affected by invasions of non-native host species. It is well structured and directly addresses the questions posed in the Introduction. The only (relatively minor) criticisms I have concern the references; these are detailed below.
1. Line 146. How could the papers by Kuchta et al. (2008) and Køie (2001) possibly help with the identifications of the cestodes and nematodes found? These papers deal with particular taxonomic groups different to those found in this study. There are other more general textbooks and reviews that would have been much more useful, especially as the identifications are only to higher taxonomic groups (an order and a family).
2. Lines 251-252. The reference to Smit & Davies (2004) is missing from the reference list.
3. Lines 275-276. Add “see” before “Diamant, 1989)” and “Merella et al., 2016)” As it stands, this implies that these are authorities for the fish names and not references.
4. I could find no text reference to the following publications given in the reference list.
Bernardi et al. (2010); El-Rashidy and Boxshall (2009, 2011 and 2014); Innocenti and Galil (2007); Innocenti et al. (2009).

Otherwise, congratulations to the authors on a useful contribution and a well-written manuscript.

---

## Round 0.2 · Minor Revisions

One of the reviewers suggested a few additional modifications. Perhaps not all of them should be followed, but I would like you to provide an answer to each suggestion.
Please pay special attention to the comments about statistics.

Reviewer 1 ·

Basic reporting

no comment

Experimental design

no comment

Validity of the findings

no comment

Additional comments

I am glad to see that the Authors have improved the quality of the article.
I attach a pdf file with the following content:
1) comments to your response
2) few (8) notes in the MS

Annotated reviews are not available for download in order to protect the identity of reviewers who chose to remain anonymous.

·

Basic reporting

.

Experimental design

.

Validity of the findings

.

Additional comments

I believe that this revised manuscript is now acceptable for publication in PeerJ without the need for a re-review.

---

## Round 0.3 · accepted · Accept

Bravo! Excellent travail!

#